# A Study on the Relationship between Depression Change Types and Suicide Ideation before and after COVID-19

**DOI:** 10.3390/healthcare10091610

**Published:** 2022-08-24

**Authors:** Sunghee Kim, Hye-Gyeong Son, Seoyoon Lee, Hayoung Park, Kyu-Hyoung Jeong

**Affiliations:** 1Interdisciplinary Graduate Program in Social Welfare Policy, Yonsei University, Seoul 03722, Korea; 2College of Nursing, Kosin University, Busan 49267, Korea; 3Institute of Symbiotic Life Technology, Yonsei University, Seoul 03722, Korea; 4Department of Social Welfare, Semyung University, Jecheon 27136, Korea

**Keywords:** COVID-19, depression, suicidal ideation, socioeconomic status, Korea

## Abstract

Background: The purpose of this study is to explore and categorize changes in depression, and investigate the relationship between suicidal ideations before and after the COVID-19 pandemic. Methods: In this study, data from the Korea Welfare Panel Study (KoWePS) was used and included 8692 adults, 19 years of age or older, who could estimate the change in depression from 2017 (12th) to 2021 (16th) for final analysis. Depression change was classified into two types, ‘low-level ascending’ type (*n* = 7809, 80.9%), and ‘increasing after reduction’ type (*n* = 883, 10.2%). The Firth Method was used to examine the relationship between depression change types and suicidal ideation. Results: The lower the equivalized annual income and the lower the educational level, and the likelihood of belonging to the ‘increasing after reduction’ type, compared to the ‘low-level ascending’ type, the greater the probability of having suicidal ideation. Conclusion: The significant impact of socioeconomic status (income and educational background) on suicidal ideation indicates the need to consider how epidemics affect inequality in society. This study is expected to provide a deeper understanding of depression, as well as to establish a foundation for long-term prevention of the rapid increase in suicide rates after COVID-19.

## 1. Introduction

Due to the rapid spread of COVID-19 infectious disease, the prolonged pandemic has had a tremendous impact on society, economy, culture, and overall daily life. In particular, new words have been coined, such as ‘COVID Blue’, which describes the negative emotions associated with the COVID-19 pandemic. Moreover, ‘Mentaldemic’ refers to the negative emotions caused by a pandemic spreading like an epidemic, and it has been argued that means of preventive action are required [1]. In fact, according to recent data on the mental health situation of Koreans, the number of people who thought of suicide during the past year due to the COVID-19 pandemic was as high as approximately 1 in 10 (8.3%). Respondents accounted for 55.8%, more than half of the population, also said that they had experienced either symptoms of depression or depression due to the COVID-19 pandemic [2]. Due to the ongoing effects of the prolonged COVID-19 pandemic, such as the rapid rise in suicide rates due to ‘COVID blue’ [3], there is a constant call for social and academic interests to be aligned in order to prevent a ‘Mentaldemic’ and maintain peoples’ good mental health.

An increase in suicide rates [4], one of the social phenomena that could occur following the COVID-19 pandemic, increases suffering and burden on families and society, ultimately lowering the country’s long-term competitiveness and hindering national development [5]. Accordingly, considering that the increase in the suicide rate has been referred to as indicative of the collapse of normality in society and the nation [6], policy efforts to reduce the rising suicide rate benefit not only individuals but also the country as a whole, and urgently need to be set as a national level. While macro and micro-level measures are constantly being explored in academia and in practice to lower the suicide rate, the major predictive factors of suicide are still difficult to find in light of the unprecedented global crisis caused by the COVID-19 pandemic.

Meanwhile, Kessler et al. [7] found that depression is the most common mental disorder among people who have experienced suicidal behavior, including suicidal ideations, and argued that depression causes suicide. Depression is characterized by persistent sadness, lethargy, a sense of worthlessness, loss of appetite, and difficulty concentrating [8]. According to the American Psychiatric Association (1994) and the Diagnostic and Statistical Manual of Mental Disorders, depression is characterized by recurrent ideations of suicide or death and attempts or plans to commit suicide [9]. A study by Hawton (2013) found that the more depression one has, the more likely one is to engage in suicide-related behaviors, such as contemplating suicide or attempting suicide [10]. In addition, about 60% suffered from major depressive disorder [11]. Taking depression, a major predictor of suicide, into account with the COVID-19 pandemic is highly significant, given previous studies show that when an individual is exposed to a pandemic like COVID-19 depression or suicidal actions can worsen [12,13].

According to recent depression-related studies, depression changes are derived through the exploration of changes in depression over time by reflecting individual heterogeneity and dynamism [14,15,16]. These studies emphasize the importance of examining the longitudinal trajectory of depression in addition to discussing high and low levels of depression measured at a particular point in time, when considering the diversity and variability of depressive symptoms. It can be described as depression changes without a single trajectory, but with multiple different trajectory types [17,18]. This implies that the types of depression changes before and after COVID-19 may vary, and the relationship between suicidal ideation may also vary depending on the types of depression changes.

Therefore, this study was conducted to explore depression before and after the COVID-19 pandemic by categorizing the changes in depression and to explore its relationship with suicidal ideation. For this, sociodemographic characteristics, including gender, age, income level, educational background, residential area, and living alone, which are known to be associated with suicidal ideation from previous studies, were included as control variables [19,20]. This study is expected to provide a deeper understanding of depression before and after the COVID-19 pandemic, as well as to establish a foundation for long-term prevention for the rapid increase in suicide rates after COVID-19.

This study aims to achieve its purpose by asking the following research questions.

What are the types of changes in depression before and after the COVID-19 pandemic?

Do changes in depression before and after the COVID-19 pandemic and suicidal ideations have any relationship to each other?

## 2. Materials & Methods

### 2.1. Data

The purpose of this study was to investigate the relationship between suicidal ideations and depression change types before and after COVID-19. In this study, the 12th–16th Korea Welfare Panel Study (KoWePS), conducted from 2017 to 2021, was used for analysis. A representative panel on Korean welfare conducted this survey in order to provide information on policies, such as the establishment of new policies and improvement of institutional structures by identifying the living conditions and welfare needs of each population group and determining the effectiveness of policy implementation. The Korean Welfare Panel Study was conducted with 8 individuals from the Korea Institute for Health and Social Affairs (KIHASA), (consisting of 1 senior researcher, 4 researchers, and 3 research assistants), and 8 individuals from the Institute of Social Welfare at Seoul National University (5 professors, 2 doctoral students, and 1 master’s student). In principle, the investigators from the Korea Institute for Health and Social Affairs directly visited the participant households, met the household members face-to-face, and recorded the responses of the respondent in CAPI (Computer Assisted Personal Interviewing) format. However, if it was difficult for the investigator to meet the participants in person during the investigation period, due to inevitable reasons, a proxy survey was conducted; for instance, in cases including returning home late at night or leaving home for a long period of time, or being absent for a certain period of time due to overseas residence, traveling or leaving for a business trip, hospital admission, military enlistment, etc. The KoWePS is conducted from February to March every year. This study included 8692 adults, 19 years of age or older, who could estimate a change in depression from 2017 (12th) to 2021 (16th), for the final analysis.

### 2.2. Variables

#### 2.2.1. Independent Variable: Depression (2017–2021)

We used the shortened Center for Epidemiologic Studies Depression Scale (CES-D) scale of Kohout et al. as an independent variable [21]. CES-D is a self-report tool that assesses depressive symptoms without a diagnosis, which was developed by Radloff and abbreviated to 11 items to alleviate the burdens of the patients by Kohout et al. [21,22]. Depression is measured on a 4-point scale on each item (1 = very rare, 2 = sometimes, 3 = often, 4 = most of the time). As the depression score increases, the level of depression is interpreted as higher. The reliability (Cronbach’s α) of the depression scale shown in this study was 0.878 in 2017, 0.881 in 2018, 0.876 in 2019, 0.865 in 2020, and 0.883 in 2021.

#### 2.2.2. Dependent Variable: Suicidal Ideation (2021)

The dependent variable was suicidal ideation, and the question was whether there had been any suicidal ideation in the past year. A value of 0 corresponded to no suicidal ideation, and a value of 1 corresponded to an ideation of suicide.

#### 2.2.3. Control Variable: Sociodemographic Characteristics (2017)

The control variables in this study were demographic and sociological characteristics; gender (male = 0, female = 1), age (continuous variable), equalized annual income (continuous variable), education level (high school or below = 0, university or above = 1), area (urban = 0, suburban and rural = 1), and living alone (living with someone = 0, living alone = 1). To calculate equalized annual income, we divided household income by the square root of household size and log-transformed for normal distribution.

### 2.3. Statistical Analysis

In this study, Stata 15.1(Stata Corp, College Station, TX, USA) and M-plus 8.0 (Muthén & Muthén, Los Angeles, CA, USA) were used to analyze data, and the analysis methods and procedures were as follows. First, descriptive statistical analysis was conducted to identify the demographic characteristics and characteristics of major variables of the analysis. Secondly, to estimate the change in depression before and after COVID-19, with the assumption that it was a single group, a potential growth model was also used. The model fit was determined by Tucker-Lewis Index (TLI), Comparative Fit Index (CFI), and Root Mean Square Error of Approximation (RMSEA), not sensitive to sample size and considering the simplicity of the model. Finally, Growth Mixture Modeling (GMM) was conducted to confirm depression type. The Growth Mixture Modeling (GMM) estimated a latent class with different effects of factors that explained individual differences within the trajectory, including the overall form of the development trajectory. Akiakie’s Information Criteria (AIC), Bayesian Information Criteria (BIC), SSABIC Sample-Size Adjusted BIC (SSABIC), Entropy, and Bootstrapped Likelihood Ratio Test (BLRT) were used to determine the optimal number of change types in the mixed growth model. Lastly, a logistic regression analysis, using the Firth Method, was used to confirm the relationship between depression change type and suicidal ideation. Using the existing logistic regression analysis, probabilities and bias parameter estimates for suicidal ideation variables might be underestimated, since only a relatively small number of individuals have suicidal ideations [23]. Thus, the existing logistic regression analysis may produce biased coefficient estimates. Therefore, this study proceeded to the penalized maximum likelihood estimation (Firth method) proposed by Firth (1993) to complement the statistical limitation [24]. The Firth Method is known be used for rare events, and to obtain unbiased coefficient estimates [25].

## 3. Results

### 3.1. Descriptive Statistics

The demographic characteristics of the study participants are shown in Table 1. There were 3575 (41.1%) males and 5117 (58.9%) females, confirming a slight majority of females. There was an average age of 52.91 years (standard deviation; SD = 16.63), and an average equivalized annual income of 22,650.76 dollars (SD = 17,843.05). As for educational background, 6154 people (70.8%) had a high school education or below, and 2538 people (29.2%) had a university degree or higher. There were 3589 people (41.3%) in urban areas compared to 5103 people (58.7%) in either suburban or rural areas. In terms of living alone, 7158 (82.4%) of those living with someone topped 1534 (17.6%) of people living alone by nearly four times. There were 158 people (1.8%) who reported having had suicidal ideation in the past year, while 8534 (98.2%) had never had suicidal ideation.

The descriptive statistical analysis of depression showed that the average depression score increased from 1.30 points in 2017 (SD = 0.41) to 1.37 points in 2021 (SD = 0.45), showing a gradual increase over time (Table 2). When the COVID-19 outbreak began in earnest in Korea in 2020 and 2021, depression increased significantly.

### 3.2. Determining Types of Depression Change

To identify overall changes in depression, a latent growth model was conducted before proceeding with the growth-mixed model (Table 3). The model fit was compared using the no growth model, the linear model, and the quadratic model. Based on the analysis, it was determined that all three models were revealed to be suitable. However, as the quadratic model (χ^2^ = 37.946 (*p* < 0.001), CFI = 0.997, TLI = 0.995, RMSEA = 0.025) met the goodness-of-fit criterion better than the other models, the quadratic model was adopted as the final model.

As a result of estimating the growth-mixed model to identify the types of depression changes based on the quadratic model, the fit of each model showed that the 4 types of classification, AIC (Akiakie’s Information Criteria), BIC (Bayesian Information Criteria), and SSABIC (Sample-Size Adjusted BIC), were found to be low compared to 1, 2, and 3 types of models (Table 4). The 2 types of classification showed that the entropy was closer to 1 than the other types. However, the 3 and 4 types contained a group that had less than 5% of the total cases, and, therefore, the 2 types of classification were finally selected.

Depression change was classified into two types, and each type was named based on its characteristics (Figure 1). In the first type, depression was at a low level overall, but it slowly rose from 2017 to 2021, so it was called the ‘low-level ascending’ type, and 7809 cases (80.9%) fell into this category. In the second type of depression, depression declined from 2017 to 2019 and then rose from 2020, and, therefore, it was classified as ‘increasing after reduction’ type, and 883 cases (10.2%) fit this category.

### 3.3. The Relationship between Depression Change Types and Suicidal Ideations

The Firth Method was used to examine the relationship between depression change types and suicidal ideation (Table 5). The study model was found to be statistically significant (χ^2^ = 139.01, *p* < 0.001). As a result of the analysis, among the control variables, equalized annual income (Coefficient; Coef. = −0.597, *p* < 0.001), educational background (Coef. = −0.539, *p* < 0.05), and the type of change in depression (Coef. = 1.511, *p* < 0.001) were found to be statistically significant determinants of suicidal ideation. In other words, the lower the equivalized annual income and the lower the educational level, and the higher the likelihood of belonging to the ‘increasing after reduction’ type, compared to the ‘low-level ascending’ type, the greater the probability of having suicidal ideation. Furthermore, gender, age, area of residence, and living alone did not significantly affect suicidal ideation.

## 4. Discussion

This study examined the relationship between depression and suicidal ideations before and after COVID-19 through classification of the types of changes among 8692 Korean adults from 2017 to 2021, including the pandemic period.

The main results of this study and its significance are as follows.

Based on longitudinal data analyzed in this study, the level of depression due to COVID-19 has been showing an upward trend. As the COVID-19 epidemic began in 2020, depression levels averaged 1.33 (SD = 0.42) in 2020, and averaged 1.37 (SD = 0.42) in 2021, while depression levels averaged from 1.30 (SD = 0.41) to 1.31 (SD = 0.42), from 2017 to 2019, showing a tendency for increase in the overall degree of depression. Furthermore, using the latent growth model and the growth mixed model to estimate these changes, the types of depression changes were classified as the ‘low-level ascending’ type (*n* = 7809, 89.8%) and ‘increasing after reduction’ type (*n* = 883, 10.2%), which indicates that the level of depression rose during the COVID-19 epidemic, from the year of 2020. The result from this study exhibited slight differences from what was observed in previous studies related to the type of depression change. The types of changes in depression observed in previous studies prior to COVID-19 usually showed various forms, such as a low level of depression, a chronic group, a decreasing or increasing group, and a group with an intermediate level of depression, implying the significance of preventing depression not only for the risk group but also for the entire population [15,16]. While there are differences in some classifications, depression type-related studies focusing mainly on the post-COVID-19 period show the results derived as a group with a low level of depression, an improved group, and a severe group [26,27]. However, as pointed out in these studies, they show limitations that have not been explored in comparing before and after COVID-19. This study examined the impact of COVID-19 by dealing with its timing, thereby displaying a new aspect of depression. The impact of COVID-19 can be seen in the upward trend in both types following the pandemic. The results of this study support previous studies showing an increase in mental health-related problems, including depression, following the spread of COVID-19 [28,29]. This was demonstrated more clearly with a longitudinal approach.

The study result examining the relationship between depression type and suicidal ideation found that suicidal ideation was more likely when depression type was ‘increasing after reduction’ compared to ‘low-level ascending’ type. Although both types of depression changes derived from this study showed an upward form after 2020, the time of the COVID-19 outbreak, it was notable that the type shown to affect suicidal ideation is an ‘increasing after reduction’ type. In the ‘increasing after reduction’ type, the depression level was higher than the ‘low-level ascending’ type before the pandemic, and it showed the characteristic of rising sharply from the period of the pandemic. In their review of the literature on post-pandemic suicidal tendencies, John et al. point out that caution should be exercised toward people with pre-existing mental health problems, such as depression, as well as people who have newly diagnosed and developed mental problems [30]. A significant finding of this study is that it confirmed a relationship between depression and suicidal ideation when depression levels are high or they are intensified by a pandemic. Among the control variables, low annual income or education was found to have a statistically significant impact on suicidal ideation. On the other hand, gender, age, area of residence, and living alone did not appear to have significance. Suicidal ideation is influenced differently by sociodemographic characteristics depending on the independent variable. There was a statistical significance between income and educational background in this study; this is consistent with previous research that showed that low income or low levels of education are highly correlated with suicidal ideation [31].

Suicidal ideations may worsen as the COVID-19 pandemic continues. The results of this study confirmed that depression is a significant risk factor for suicidal ideation. In particular, the continuous rise in the level of depression, not only in 2020, right after the outbreak of COVID-19, but also in 2021, shows the need to prepare for the long-term impact of COVID-19 on mental health. In addition, the ‘increasing after reduction’ type, which has been shown to have a positive association with suicidal ideation, displayed a relatively high level of depression even before COVID-19. In light of this, it is likely that the individuals in this group had previously suffered from mental health disorders and were subjected to greater difficulty as a consequence of COVID-19. In fact, some previous studies have reported that people who have previously suffered from mental health problems experienced high levels of mental health issues after COVID-19 [27,30]. However, this cannot rule out the possibility that prior to COVID-19, the individuals may not have been aware of the level of their depression. Since the ‘increasing after reduction’ type showed a sharp rise in the level of depression after COVID-19, attention should also be paid to those who have suffered a blow to their mental health due to rapid difficulties from COVID-19. Lastly, the significant impact of socioeconomic status (income and educational background) on suicidal ideation indicates the need to consider how disasters, such as COVID-19, affect inequality in society, which shows the significance of practical intervention.

## 5. Conclusions

According to the findings of this study, interventions are needed to counter depression caused by COVID-19, while preparations should also be made for long-term effects on mental health. Several studies have refuted the myth that COVID-19 negatively affects mental health [32]. There is a need, however, for more longitudinal studies from more diverse countries to refute this claim [33]. This study is meaningful in that it examined the mental health of Koreans longitudinally before and after the COVID-19 pandemic, and the impact of COVID-19. If further research can reveal in detail factors that influence depression or methods that mediate the mechanism of depression on suicidal ideation in the future, then countermeasures can be discussed in more detail. The data used in this study is the largest panel data in Korea, however, additional research is recommended due to the high proportion of low-income groups within the data. In addition, there is a limitation in absence of the variables related to the clinical characteristics associated with suicidal ideation that were not considered as control variables. Therefore, it is necessary to utilize clinical characteristic variables in follow-up studies.

## Figures and Tables

**Figure 1 healthcare-10-01610-f001:**
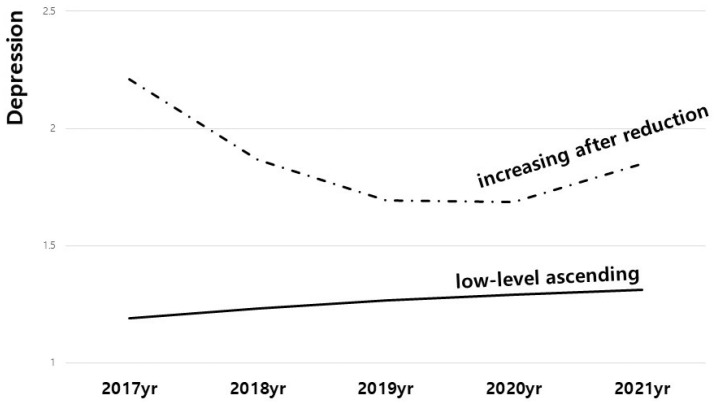
Estimation of Depression Change Types.

**Table 1 healthcare-10-01610-t001:** Sociodemographic characteristics of the participants (*N* = 8692).

Variable	Categories	*N*	%
Gender	Male	3575	41.1
	Female	5117	58.9
Age (M(SD))		52.91 (16.63)	
$ Equalized Annual Income (M (SD))		22,650.76 (17,843.05)	
Education level	High school or below	6154	70.8
	University or above	2538	29.2
Area of living	Urban	3589	41.3
	Suburban and rural	5103	58.7
Living alone	No	7158	82.4
	Yes	1534	17.6
Suicidal ideation over past year	No	8534	98.2
	Yes	158	1.8

Note: M = Mean; SD = Standard Deviation; $ = USD.

**Table 2 healthcare-10-01610-t002:** Descriptive statistics of depression (*N* = 8692).

Variable	Min	Max	M	SD
Depression—2017	1.00	4.00	1.30	0.41
Depression—2018	1.00	3.91	1.30	0.41
Depression—2019	1.00	3.91	1.31	0.42
Depression—2020	1.00	3.82	1.33	0.42
Depression—2021	1.00	4.00	1.37	0.45

Note: Min = Minimum; Max = Maximum; M = Mean; SD = Standard Deviation.

**Table 3 healthcare-10-01610-t003:** Model fit for Latent Growth Model for Depression Change.

Model	χ^2^	CFI	TLI	RMSEA
No growth model	823.614 ***	0.909	0.925	0.085
Linear model	229.499 ***	0.961	0.961	0.050
Quadratic model	37.946 ***	0.997	0.995	0.025

Note: CFI = Comparative Fit Index; TLI = Tucker-Lewis Index; RMSEA = Root Mean Square Error of Approximation. *** *p* < 0.001.

**Table 4 healthcare-10-01610-t004:** Model fit of Growth Mixture Modeling. (*N* = 8692).

Class	Model Fit					Groups
	AIC	BIC	SSABIC	Entropy	BLRT *p*-Value	n (%)
1	36,575.413	36,674.395	36,629.905	-	-	-
2	33,474.931	33,602.194	33,544.993	0.927	<0.001	7809 (89.8), 883 (10.2)
3	32,302.587	32,415.710	32,364.865	0.909	<0.001	7449 (85.7), 845 (9.7), 398 (4.6)
4	31,629.011	31,763.344	31,702.965	0.895	<0.001	7193 (82.8), 715 (8.2), 398 (4.6), 386 (4.4)

Note: AIC = Akiakie’s Information Criteria; BIC = Bayesian Information Criteria; SSABIC = Sample-Size Adjusted BIC; BLRT = Bootstrapped Likelihood Ratio Test.

**Table 5 healthcare-10-01610-t005:** The Relationship Between Depression Change Types and Suicidal Ideations.

Variables	Coef.	S.E.	OR
Gender (ref. male)	0.062	0.179	1.063
Age	−0.008	0.007	0.992
Equalized Annual Income (log)	−0.597 ***	0.144	0.550
Education	−0.539 *	0.267	0.583
Area (ref. urban)	−0.245	0.164	0.783
Living alone (ref. Living with someone)	−0.260	0.210	0.771
Depression change type (ref. low-level ascending)	1.511 ***	0.181	4.532
constant	2.116	1.591	8.296

Note: Coef. = Coefficient; SE = Standard Error; OR = Odds Ratio; ref = reference. * *p* < 0.05, *** *p* < 0.001.

## Data Availability

The datasets generated during and/or analyzed in this study are publicly available upon request from: https://www.koweps.re.kr:442/eng/data/data/list.do. (accessed on 10 May 2022).

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
