# Peer review of "A Study on the Relationship between Depression Change Types and Suicide Ideation before and after COVID-19"

_healthcare, 2022, doi:10.3390/healthcare10091610_

Round 1

Reviewer 1 Report

Dear authors,
Thank you for your paper named "A Study on the Relationship Between Depression Change Types and Suicide Ideation Before and After COVID-19." The topic is interesting considering the evolving situation of the COVID-19 pandemic and the large sample assessed in this study. However, some issues should be addressed to improve the paper.

1) Please refrain from interpreting the results as a "causal relationship". In particular, on line 226-227, P7. Even though this study used longitudinal data with large sample size, it's still an observation study. Generally, observational data are subject to biases from confounding, selection and measurement, which can result in an underestimate or overestimate of the effect of interest. Even though sometimes causal inferences can be drawn from observational studies, as long as certain conditions are met, I don't think this study has addressed all the necessary confounding variables. Specifically, no clinical characteristics were included as controlled variables. Can the authors justify not including any clinical characteristics? For example, it is highly possible that those individuals with more comorbidity may be more mentally vulnerable during the pandemic. There are other clinical characteristics like self-rated health that are highly related to suicide ideation and you can't just ignore them.

2) In the introduction, can the authors cite other studies that categorize depression? And ideally, compare the resulted 2-type category in this study to previous studies in the discussion section. One important factor to consider when determining the optimal number of changes types is substantive interpretability. So more comments on the selection is recommended.

Author Response

1) Please refrain from interpreting the results as a "causal relationship". In particular, on line 226-227, P7. Even though this study used longitudinal data with large sample size, it's still an observation study. Generally, observational data are subject to biases from confounding, selection and measurement, which can result in an underestimate or overestimate of the effect of interest. Even though sometimes causal inferences can be drawn from observational studies, as long as certain conditions are met, I don't think this study has addressed all the necessary confounding variables. Specifically, no clinical characteristics were included as controlled variables. Can the authors justify not including any clinical characteristics? For example, it is highly possible that those individuals with more comorbidity may be more mentally vulnerable during the pandemic. There are other clinical characteristics like self-rated health that are highly related to suicide ideation and you can't just ignore them.

→ I appreciate your comment.  As you mentioned in your comment, we acknowledge that referring to the results as a "casual relationship" should be refrained from. Therefore, the word ‘causal’ was deleted and only ‘relationship’ was written (page 8, line 259).

In addition, we agree that it is essential to consider the control variables including clinical characteristics; however, due to the limitations of the secondary data, it was not possible to utilize these variables. Therefore, as a limitation of this study, we stated under conclusion (page 9, line 301-304) that the control variables including clinical characteristics could not be considered, it is necessary to utilize clinical characteristics variables in follow-up studies.

2) In the introduction, can the authors cite other studies that categorize depression? And ideally, compare the resulted 2-type category in this study to previous studies in the discussion section. One important factor to consider when determining the optimal number of changes types is substantive interpretability. So more comments on the selection is recommended

→ In response to your feedback, we added references to existing studies looking at the types of depressive changes in the introduction (page 2, line 70-72). Additionally, in the discussion section, the results were compared with previous studies that examined depression changes (page 8, line 232-243). As a result of these detailed statements, this paper better describes the findings of this study, which examined depression types during the COVID-19 period.

Reviewer 2 Report

Dear authors, 

The manuscript tries to find the changes in depression before and after covid-19 pandemic. I have some considerations to your work.

-          Introduction: quite good and complete.

-          Methodology:

Data at the variable section: why data are different? One of them has been collected at 2021 and the other at 2017?

Living alone: people at residences have been taking into account?

Maybe the employment situation should be considered.

There is a lack of information about the procedure, how participants were contacted? When the survey was fill in? It was face to face, via telephone? or maybe I think that there is a retrospective study, which sources have been accessed to find the information without contact patients?

How many persons are involved in this research? How did the share the work, all of them participate in all steps of the investigation?

-          Results:

Are the same people evaluated during these years?

Line 172: abbreviations should be explained the first time you use them.

All tables must have an abbreviations explanation after them.

-          Discussion

More literature comparing your results should be added.

-          References

Please, revise mdpi rules about how to cite to adapt your references.

Author Response

Introduction: quite good and complete. →Thank you.

-          Methodology:

Data at the variable section: why data are different? One of them has been collected at 2021 and the other at 2017? →Since the control variable, demographic characteristics, and the independent variable, depression, should precede suicidal ideation, the dependent variable, in time, we used 2017 data. In addition, since this study is to examine depression types of change over time, we used data from 2017 to 2021.

Living alone: people at residences have been taking into account? → Yes. The living alone variable was used based on the number of people living together.

Maybe the employment situation should be considered. → Thank you for your good recommendation. However, despite the existence of the employment status variable in the data, it was excluded for analysis due to the annual income variable that we have included in the manuscript and multicollinearity issues.

There is a lack of information about the procedure, how participants were contacted? When the survey was fill in? It was face to face, via telephone? or maybe I think that there is a retrospective study, which sources have been accessed to find the information without contact patients?

→ Thank you for your suggestion. The procedure, method, and duration are stated under 2.1.data section (page 3, line 103-112). In principle, the investigators from the Korea Institute for Health and Social Affairs directly visit the participant households, meet the household members face-to-face, and record the responses of the respondent in CAPI (Computer Assisted Personal Interviewing) format. However, if it is difficult for the investigator to meet the participants in person during the investigation period due to inevitable reasons, the proxy survey was conducted; the cases include returning home late at night or leaving home for a long period of time, or being absent for a certain period of time due to overseas residence, traveling or leaving for a business trip, hospital admission, military enlistment, etc. The KoWePS has been conducted from February to March every year.

How many persons are involved in this research? How did the share the work, all of them participate in all steps of the investigation?

→ Thank you for your suggestion. The number of persons involved in this research is included 2.1.data section (page 3, line 99-103). The Korean Welfare Panel Study was conducted with 8 individuals from the Korea Institute for Health and Social Affairs (KIHASA), (consisting of 1 senior researcher, 4 researchers, and 3 research assistants), and 8 individuals from the Institute of Social Welfare at Seoul National University (5 professors, 2 doctoral students, and 1 master's student).

-          Results:

Are the same people evaluated during these years? → Yes. The same people are evaluated during the years.

Line 172: abbreviations should be explained the first time you use them. → Yes. I have made changes in explaining the abbreviations in the manuscript accordingly.

All tables must have an abbreviations explanation after them.. → Yes. I have made changes in explaining the abbreviations under all the tables.

-          Discussion

More literature comparing your results should be added. → Some paragraphs have been moved to discussion, and references have been cited to facilitate a richer discussion.

-          References

Please, revise mdpi rules about how to cite to adapt your references. → Thank you for pointing out an important point. I have made changes in the reference according to the mdpi rules.

Reviewer 3 Report

Overall, the manuscript is well written and reads well. However, stylistic errors should be checked throughout the manuscript.

Since this study is conducted with the data from the Korea Welfare Panel Study, the tile should indicate the Korean context so that readers can recognize the context of the study from the beginning.

With regard to the method of this study, it would be desirable to explain the method as fully and in detail as possible since some readers may not be familiar with this type of method.

In the conclusion section, the authors provided future directions. However, I guess it would be required for authors to provide some implications for the Korean society and the world as well if possible.

Thank you. 

Author Response

Since this study is conducted with the data from the Korea Welfare Panel Study, the tile should indicate the Korean context so that readers can recognize the context of the study from the beginning.

→ Thank you for the good comments. A detailed explanation of the Korea Welfare Panel Study was presented in the method part.

With regard to the method of this study, it would be desirable to explain the method as fully and in detail as possible since some readers may not be familiar with this type of method. →  Under '2.3. Statistical analysis', we have put an additional explanation about Growth Mixture Modeling (GMM) more specifically (page 4, line 145-147)

In the conclusion section, the authors provided future directions. However, I guess it would be required for authors to provide some implications for the Korean society and the world as well if possible.

→ The significance of this study is additionally presented in the conclusion section (page 9, line 293-296).

Round 2

Reviewer 2 Report

Dear authors, 

I agree with the changes made.

Line 117, 119, 256: years should be deleted.

References: follow mdpi rules.

Thank you.

Author Response

I agree with the changes made.

→ Thank you for your helpful comments.

Line 117, 119, 256: years should be deleted.

→ We appreciate your suggestion. We have made changes accordingly.

References: follow mdpi rules.

→ Thank you for your review. We have downloaded mdpi endnote and modified the reference accordingly.
